# The KEAP1-NRF2 System in Healthy Aging and Longevity

**DOI:** 10.3390/antiox10121929

**Published:** 2021-11-30

**Authors:** Daisuke Matsumaru, Hozumi Motohashi

**Affiliations:** 1Laboratory of Hygienic Chemistry and Molecular Toxicology, Gifu Pharmaceutical University, 1-25-4 Daigaku-Nishi, Gifu 501-1196, Japan; matsumaru-da@gifu-pu.ac.jp; 2Department of Gene Expression Regulation, Institute of Development, Aging and Cancer, Tohoku University, 4-1 Seiryo-cho, Aoba-ku, Sendai 980-8575, Japan

**Keywords:** KEAP1-NRF2 system, oxidative stress, aging, longevity, cell senescence, tissue aging, age-related hearing loss, Alzheimer’s disease, sarcopenia

## Abstract

Aging is inevitable, but the inherently and genetically programmed aging process is markedly influenced by environmental factors. All organisms are constantly exposed to various stresses, either exogenous or endogenous, throughout their lives, and the quality and quantity of the stresses generate diverse impacts on the organismal aging process. In the current oxygenic atmosphere on earth, oxidative stress caused by reactive oxygen species is one of the most common and critical environmental factors for life. The Kelch-like ECH-associated protein 1-NFE2-related factor 2 (KEAP1-NRF2) system is a critical defense mechanism of cells and organisms in response to redox perturbations. In the presence of oxidative and electrophilic insults, the thiol moieties of cysteine in KEAP1 are modified, and consequently NRF2 activates its target genes for detoxification and cytoprotection. A number of studies have clarified the contributions of the KEAP1-NRF2 system to the prevention and attenuation of physiological aging and aging-related diseases. Accumulating knowledge to control stress-induced damage may provide a clue for extending healthspan and treating aging-related diseases. In this review, we focus on the relationships between oxidative stress and aging-related alterations in the sensory, glandular, muscular, and central nervous systems and the roles of the KEAP1-NRF2 system in aging processes.

## 1. Introduction

Living organisms on this planet are exposed to oxygen, sunlight, and various chemicals in the atmosphere, soil, and water. In addition to these exogenous environmental factors, endogenously produced chemicals and metabolites often perturb cellular and organismal functions. To cope with such perturbations, we are all equipped with defense mechanisms, each of which specializes in an individual stress and continuously responds to the stress for adaptation and maintenance of homeostasis. In response to continuous stresses throughout life, maladapted cell populations and their unrepaired damage gradually accumulate, resulting in the functional decline of tissues and organs in aged organisms. One of the most common stresses that impact the aging process is oxidative stress. It is commonly accepted that molecular and cellular damage resulting from reactive oxygen species (ROS) or oxidative stress accelerates the aging process [1]. This oxidative stress theory of aging is the most popular explanation for the molecular mechanisms of aging among a number of theories that have been proposed [2]. It explains many aging phenotypes at the molecular level, including failure of mitochondrial integrity, proteostasis, and barrier structure as well as the decline of DNA repair, immune function, and regenerative capacity [3]. Although the oxidative stress theory of aging is widely accepted, it has been challenged by several caveats. Some long-living species exhibit high levels of oxidative damage even at young ages [4], and increased levels of antioxidants have failed to prolong longevity in several cases (reviewed in [5]). Moreover, non-toxic levels of ROS function as signaling molecules that induce protective defense in responses to age-dependent damage [6]. Therefore, controlling and adjusting redox balance in appropriate ways according to the various cellular contexts is likely to be necessary for the enhancement of our health.

Nuclear factor erythroid-derived 2-like 2 (NRF2; encoded by the *Nfe2l2* gene) is a member of the cap‘n’collar (CNC) protein family and coordinately regulates a battery of cytoprotective genes. Under unstressed conditions, NRF2 is bound by Kelch-like-ECH-associated protein 1 (KEAP1) in the cytoplasm and is constantly ubiquitinated for degradation by proteasomes (Figure 1) [7,8,9]. When cells are exposed to ROS and electrophiles, the thiols of cysteine residues in KEAP1 are directly modified, leading to decreased KEAP1-dependent ubiquitination of NRF2 and rapid accumulation of newly synthesized NRF2. Subsequently, stabilized NRF2 translocates to the nucleus and forms a heterodimer with small musculo-aponeurotic fibrosarcoma (sMAF) proteins, inducing transcriptional activation by binding to antioxidant-responsive elements (AREs) [10,11,12,13] or electrophile-responsive elements (EpREs) [14]. Canonical NRF2 target genes encode factors required for glutathione synthesis (*Gclc* and *Gclm*), detoxifying ROS and xenobiotics (*Txnrd1*, *Prdx1* and *Nqo1*), heme metabolism (*Hmox1*), phase II conjugation, DNA repair, NADPH production, proteostasis, and so on (Figure 2; reviewed in [15,16,17]).

In addition to elimination of ROS, ARE-independent transcriptional interference by NRF2 has been reported and contributes to anti-inflammatory functions. Induction of proinflammatory cytokine genes, such as *Il6*, and murine inflammatory phenotype models, including experimental autoimmune encephalomyelitis (EAE) and *Staphylococcus aureus* infection models, were ameliorated by supplementation with chemical NRF2 inducers and genetic activation of *Nrf2* [18]. It was also found that systemic activation of NRF2 by *Keap1* knockdown ameliorated tissue inflammation and lethality in *Scurfy* mice, which are deficient in regulatory T cells [19]. Similarly, it is expected that NRF2 contributes to the amelioration of chronic smoldering inflammation under both physiological and pathological conditions.

Although activation of the KEAP1-NRF2 system reduces the expression of proinflammatory cytokine genes, its constitutive activation by *Nrf2* gain-of-function mutation or *Keap1* mutation may represent a risk to maintaining physiologically healthy conditions. For instance, constitutive activation of NRF2 resulted in reduced quiescence of long-term hematopoietic stem cells in steady-state hematopoiesis [20], attenuated differentiation of both osteoclasts and osteoblasts [21], severe hyperkeratosis of the esophagus and forestomach in the juvenile [22], and therapeutic resistance and aggressive tumorigenic activity in cancer cells [23,24,25,26]. These observations suggest that transient activation of the KEAP1-NRF2 system is beneficial but that persistent activation is not.

In this review, the contribution of the KEAP1-NRF2 system to aging-related conditions and diseases is described, including cellular senescence and organismal aging/longevity (Figure 2). In addition, the possibility of intervening in the aging process by modulating the KEAP1-NRF2 system is discussed.

## 2. Cellular Senescence and the KEAP1-NRF2 System

Oxidative stress increases during aging. As part of the DNA damage response, DNA damage foci are formed and significantly increase with age in the lung, spleen, dermis, liver, and gut epithelium [27]. Such DNA damage is a major trigger of cellular senescence, which is one of nine defined hallmarks of aging [28]. Cellular senescence is a cell state implicated in various physiological processes and a wide spectrum of age-related diseases [29]. In addition to DNA damage, exposure to chemotherapeutic drugs, oxidative stress, mitochondrial dysfunction, and oncogene activation can cause cellular senescence (Figure 3). Cellular senescence has been considered beneficial, for example, for contributing to the clearance of damaged and potentially oncogenic cells from tissues. Senescent cells secrete proinflammatory cytokines and matrix metalloproteinases, referred to as the senescence-associated secretory phenotype (SASP) [30,31]. This phenotype worsens inflammation and disease conditions. Selective removal of senescent cells by small compounds or chimeric T cells has been shown to be beneficial for improving pathologies of age-associated diseases and for extending lifespan [32,33,34]. The relationship between cellular senescence and the KEAP1-NRF2 system has been investigated (Figure 3). In some occasions, NRF2 signaling decreases with aging due to downregulation of NRF2 expression and transcriptional activity [35,36]. NRF2 activity declines during senescence, whereas silencing NRF2 leads to premature senescence, implying a negative spiral of NRF2 dysfunction and cell senescence [37]. Consistently, genetic depletion of *Nrf2* enhances age-related induction of senescence markers and inflammatory SASP factors, exacerbating the inflammatory status of the hippocampus [38]. Activation of the KEAP1-NRF2 system is expected to suppress smoldering inflammation and to attenuate physiological dysfunction during aging. Transient pharmacological activation of NRF2 in endothelial progenitor cells from aged mice protected these cells against oxidative stress, ameliorated their biological dysfunction and downregulated the NLR family pyrin domain containing 3 (NLRP3) inflammasome [39]. In contrast, persistent genetic activation of NFR2 in skin fibroblasts induces cellular senescence and leads to a cancer-associated fibroblast phenotype through regulation of the matrisome [40]. Here, again, transient activation of NRF2 is beneficial, whereas persistent activation of NRF2 is often detrimental, potentially explaining why NRF2 is so tightly regulated at multiple levels from gene expression [41] to transcript stability [42] to protein stability [8].

## 3. Longevity and the KEAP1-NRF2 System

One of the central topics in aging research is the factors affecting longevity among species. Historically, many researchers have discussed the correlation between longevity and body size and observed a tendency for a proportional relationship between them. However, several species possess much longer, or shorter, longevity than expected. In particular, Brandt’s bats and naked mole-rats show much longer lifespans than that expected based on their body size [4,43]. In the case of humans, the natural lifespan is estimated to be approximately 30 years, but it is approximately 80 years in most developed countries [44].

A number of genetically modified mouse models exhibit increased longevity (reviewed in [45]). Such experimental models and long-lived species are resistant to both endogenous and environmental stressors and resist age-related diseases such as cardiovascular and neurodegenerative diseases and cancers [46]. Although excess ROS reduce lifespan by causing extensive cellular dysfunction and damage, birds are remarkably long-lived. Generally, cellular stress resistance is an evolutionarily conserved feature of longevity [47]. The KEAP1-NRF2 system is one of the major mechanisms that enhances cellular stress resistance. Constitutive activation of NRF2 has been observed in ~95% of bird species, representing an adaptive mechanism capable of counterbalancing high ROS levels [48]. In rodents, comparative analysis of naked mole-rats and nine other rodent species revealed a positive correlation between lifespan and NRF2 activity. This observation was verified by a negative correlation between lifespan and suppressors of NRF2, i.e., KEAP1 and βTrCP, which are involved in the degradation of NRF2 [46]. In male fruit flies, *keap1* loss-of-function mutations have significantly beneficial effects on oxidative stress tolerance and longevity [49,50]. In worms, constitutive nuclear accumulation of SKN-1, an ortholog of Nrf/CNC proteins, increases stress tolerance and longevity [51].

Although increased stress tolerance and longevity seem to be closely related, they are not necessarily equal (Figure 4). SKN-1/NRF2 deficiency results in increased vulnerability to oxidative stress and a shortened lifespan in worms. The latter is rescued by DAF-16/FoxO overexpression, which is related to the insulin/IGF-1 signaling pathway, but the former is not, implying that the mechanisms underlying resistance to oxidative stress and longevity are distinct [52]. Another example has been shown in a fly study. While mild NRF2 activation extends lifespan, induction of NRF2 activation at high levels in adult flies results in accelerated aging accompanied by signs of type 1 diabetes with altered mitochondrial bioenergetics [53]. There seems to be a trade-off between extreme stress tolerance and aging acceleration (Figure 4).

A similar trade-off is observed in the emergence of cancer cells with persistent activation of NRF2. Loss-of-function of *Keap1* or gain-of-function of *Nrf2* due to somatic mutations in their respective genes is frequently observed in solid tumors that occur in the lung, head and neck, and bladder [54,55,56]. Consequent persistent activation of NRF2 in cancer cells results in therapeutic resistance [57,58]. Such cancer cells are highly dependent on NRF2 activity for their survival and proliferation, and this status is designated NRF2 addiction [23,25,59]. The most characteristic feature of NRF2-addicted cancer cells is their extremely enhanced detoxification and antioxidant capacities based on the massive production of glutathione and massive uptake of cystine via the cystine transporter xCT, which is a cystine/glutamate antiporter [60]. Because glutamate is excreted via xCT and consumed for glutathione synthesis, the robust stress tolerance of NRF2-addicted cancer cells is thought to occur at the cost of metabolic imbalances, which needs to be corrected by additional supplementation with glutamate [61].

Although many antioxidant drugs have failed to modify the mammalian lifespan [5,62], it has been reported that treatment with NRF2-inducing agents exerts favorable effects. Protandim, a mixture of botanical extracts including bacosides, silymarin, withaferin A, epigallocatechin-3-gallate, and curcumin, activates NRF2 and extends median lifespan in male mice [63]. In fruit flies, lithium extends lifespan when administered throughout adulthood or even only later in life by inhibiting glycogen synthase kinase-3 (GSK-3), resulting in consequent activation of NRF2. Intriguingly, combining genetic loss of *Keap1* with lithium treatment revealed that high levels of NRF2 activation conferred stress resistance, while low levels additionally promoted longevity [64], consistent with the trade-off paradigm discussed above.

## 4. Tissue Aging and the KEAP1-NRF2 System

Judging from the distribution of cells positive for senescence-associated β-Gal and oxidative stress markers, aging does not occur in a uniform manner among tissues in an organism. In this review, we focus on sensory systems, glandular structures, the central nervous system, and skeletal muscles as organs with aging processes that can be modified by activation of the KEAP1-NRF2 system (Figure 2 and Figure 5).

### 4.1. Aging in Sensory Organs and the KEAP1-NRF2 System

Age-related hearing loss (AHL), also known as presbycusis, is the most common type of sensorineural hearing loss in the elderly [65]. It is characterized by degenerative and irreversible changes in inner ear sensory cells (Figure 5A) [66]. Histologically, impairment has been reported in hair cells, spiral ganglion neurons, spiral ligament, and stria vascularis [67,68]. Various factors causing AHL have been reported, such as ROS [69], exposure to noise [70,71], ototoxic chemicals [72], systemic diseases [73,74], and genetic predispositions [75]. Most of these factors are more or less related to oxidative stress when they damage cells. Excessive oxidative stress and/or decreased antioxidant capacity induces oxidative damage in the cochlea [76,77,78]. During the pathogenesis of AHL, the contribution of inflammation has also been described, as in the case of noise-induced hearing loss, which is another major class of sensorineural hearing loss [79,80].

The KEAP1-NRF2 system protects cochlear cells from oxidative stress and inflammation and contributes to the avoidance of hearing loss. The C57BL/6 mouse strain is a well-studied model of early-onset AHL with a SNP in the *Cdh23* gene [81]. A decline in hearing first becomes apparent at high frequencies as early as 3–6 months of age [82] and progresses to severe impairment by one year of age [83], which corresponds to middle age in C57BL/6 mice [84]. NRF2 is expressed in the inner and outer hair cells and supporting cells of the organ of Corti throughout the cochlea and is decreased in the organ of Corti in older individuals [85]. Its suggested contribution to cytoprotection has been demonstrated in genetically modified mice. Although *Nrf2^–/–^* mice maintained normal auditory thresholds at 3 months of age, their cochlear structure and function were significantly deteriorated compared to those of age-matched wild type mice at 11 months of age [86]. This result indicates that endogenous NRF2 is essential for resisting the progression of age-related pathology in the auditory system. In contrast, genetic NRF2 activation achieved by *Keap1* knockdown (*Keap1*-KD) in mice enhanced the expression of multiple NRF2 target genes, ameliorated cochlear degeneration, and maintained hearing ability at 12 months of age compared to those in wild type mice [87]. Similarly, noise-induced hearing loss was exacerbated in *Nrf2^–/–^* mice and prevented by pretreatment with the NRF2 inducer 2-cyano-3,12-dioxooleana-1,9-dien-28-imidazolide (CDDO-Im) [88]. These results indicate that suppression of oxidative stress by NRF2 activation contributes to the alleviation of age-related structural alterations and functional decline in the cochlea. Indeed, many reports have shown that drugs activating the KEAP1-NRF2 system are beneficial for hearing protection in vitro and in vivo (reviewed in [89]). NRF2 activation is likely to be a general strategy for inner ear protection. In addition to inner ear, aging related eye disease such as age-related macular degeneration is also caused by oxidative stress-induced damage to the retinal pigment epithelium and can be ameliorated by genetic activating NRF2 [90].

### 4.2. Aging in Glandular Structures and the KEAP1-NRF2 System

Among organs and tissues, glandular structures are essential for retaining quality of life. Aging is a risk factor for dry eye disease, which is a status of functional decline of the lacrimal gland [91]. Oxidative stress is suggested to be a causative factor for the pathogenesis of dry eye disease [92]. The lacrimal system consists of the lacrimal glands, the tear film in contact with the conjunctiva and cornea, and the lacrimal drainage system through the nasolacrimal duct to the nose [93]. Lacrimal glands undergo structural and functional alterations with increasing age, and an increase in oxidative stress may play roles in the decline of lacrimal gland function with age (Figure 5B). Age-related morphological changes in lacrimal glands include diffuse fibrosis, diffuse atrophy, and periductal fibrosis, which may be related to the decrease in tear outflow with age and interlobular ductal dilatation [94].

Critical roles of NRF2 in cytoprotection and anti-inflammation in the lacrimal system have been reported [95,96,97]. In addition, the antiaging effects of NRF2 in the lacrimal gland have also been described. In the lacrimal gland of aged mice, ROS accumulation and heavy infiltration of mononuclear cells are evident [98]. When Oltipraz, an NRF2 inducer, was administered to aged mice, oxidative stress markers such as nitrotyrosine and 4-hydroxy-2-nonenal (4-HNE) were decreased in the lacrimal gland. Concomitantly, infiltration of immune cells into the lacrimal gland was also decreased, which was accompanied by a significant increase in conjunctival goblet cell density compared to aged mice fed a standard diet [99].

Dry mouth (salivary hypofunction or xerostomia) is another common complaint among aged people, often resulting in oral diseases such as dental caries and periodontal disease that is associated with chewing, swallowing, and speaking difficulties. In addition to aging, xerostomia is also caused by medication, high doses of radiation, certain diseases such as Sjögren’s syndrome, and so on. The aging process is associated with reduced salivary flow in a salivary gland-specific manner [100]. Saliva seems to undergo chemical changes with aging. As the amount of ptyalin decreases and mucin increases, saliva becomes thick and viscous and presents problems for the elderly [101]. Histological analysis has revealed an age-related decrease in the proportion of parenchymal tissue versus stromal tissue in salivary glands [102,103]. Once again, oxidative stress is an important factor in understanding the aging phenotypes of salivary glands (Figure 5B). Hyposalivation and structural changes, parenchymal atrophy, fatty degeneration, and stromal fibrosis are coupled with a reduction in the antioxidant capacity of salivary glands in aged mice [104].

Similar to its roles in the lacrimal gland, NRF2 contributes to cytoprotection and anti-inflammation in the salivary gland. Intense periductal lymphocyte infiltration is observed in the salivary glands of *Nrf2^–/–^* mice [95]. The antiaging function of NRF2 in the salivary gland has been demonstrated in *Keap1*-KD mice. Aging phenotypes of the salivary gland, such as iron and collagen deposition, immune cell infiltration, increased DNA damage and apoptosis accompanied by elevated oxidative stress, are all markedly attenuated in *Keap1*-KD mice [105]. Intriguingly, anethole trithione, which has been shown to increase salivary flow and is clinically used for the treatment of hyposalivation [106], induces the expression of NRF2-dependent genes [107]. Treatment with astaxanthin, which possesses strong antioxidant and anti-inflammatory effects [108], also prevents age-related hyposalivation and inflammation in mice [109]. These observations suggest that enhancing both antioxidant and anti-inflammatory functions simultaneously is essential for maintaining healthy salivary glands and for the prevention of hyposalivation in the elderly.

### 4.3. Aging in the Brain, Neurodegenerative Diseases and the KEAP1-NRF2 System

Brain aging is a critical and common factor underlying neurodegenerative diseases and dementia [110]. The brain shrinks with increasing age and suffers from deteriorating changes at the molecular, cellular, tissue, and functional levels [111]. Similar to other organs, oxidation of biomolecules, such as protein carbonylation and oxidized nucleic acids, increases in an age-dependent manner [112]. Within a physiological range of alterations, age-related memory impairment has been shown to correlate with antioxidant capacities. For example, plasma antioxidant vitamin levels correlate with cognitive performance in healthy older people [113]. Increased levels of oxidative stress and/or antioxidant deficiencies are suggested to be risk factors for cognitive decline [114]. Intracellular glutathione concentrations decrease with age in the mammalian brain, especially in the hippocampus [115]. Under pathological conditions, oxidative stress has been implicated in the progression of a number of neurodegenerative diseases, including Alzheimer’s disease (AD), Parkinson’s disease (PD), and amyotrophic lateral sclerosis (ALS) (Figure 5C) [116]. Oxidative stress and inflammation are increased in the brains of AD patients, which is widely recapitulated in a number of model animals [117,118,119]. Decreases in antioxidant molecules, including glutathione, glutathione peroxidase, glutathione-S-transferase, and superoxide dismutase, have been observed in mitochondrial and synaptosomal fractions of the postmortem frontal cortex derived from individuals with mild cognitive impairment and AD patients [120]. Low levels of endogenous antioxidants and increased reactive species have also been described in PD [121]. In addition to reduced antioxidant capacities, a number of reports have described the so-called neuroinflammatory status in AD and PD models and patients [122,123,124,125].

As in other organs, the KEAP1-NRF2 system plays important roles in the maintenance of brain function [126,127,128]. Although *Nrf2* is expressed in neurons, astrocytes, and microglial cells, it is substantially more active in astrocytes and microglial cells rather than neurons [129,130]. NRF2 strongly enhances glutathione synthesis in the brain, especially in astrocytes [10,127]. Glutathione produced in astrocytes is transported to neurons and exerts beneficial effects in protecting neurons from oxidative damage [127,131].

In the aging brain, mRNA and protein expression levels of NRF2 appear to be decreased in general but increased at specific regions due to an adaptive response to pathological changes [132,133]. The NRF2 activities are also altered in the brains of AD patients and AD model *App^NL-G-F/NL-G-F^* knock-in mice [134,135]. NRF2 deficiency aggravates phenotypes of AD model mice, such as *APP/TAU* mice and *APP/PS1* mice [136,137,138,139]. Conversely, genetic NRF2 activation by *Keap1* knockdown in *App^NL-G-F/NL-G-F^* knock-in mice represses inflammatory cytokine gene expression, enhances glutathione synthesis, and reverses memory impairment [140]. Similarly, overexpression of *Nrf2* by viral vectors protects hippocampal neurons of *APP/PS1* mice and cultured hippocampal cells [141,142]. Pharmacological approaches to induce NRF2 activation have been performed to ameliorate neurodegenerative diseases [133]. The NRF2-activating chemicals CDDO-methyl-amide and dimethyl fumarate (DMF) have been shown to improve cognitive function in AD model mice [138,143]. Mild, long-term pharmacological induction of NRF2 using 6-(methylsulfinyl)hexyl isothiocyanate (6-MSITC) suppresses AD-like pathology in *App^NL-G-F/NL-G-F^* knock-in mice [140]. The beneficial effects of NRF2 have also been reported in pathological status of PD. Dysregulation of the KEAP1-NRF2 system has been described in PD [144,145]. In an MPTP-induced PD mouse model, *Nrf2* deficiency exacerbates astrogliosis and microgliosis with elevated expression of inflammation markers [146]. Treatment with 6-MSITC protects neuronal functions in PD model mice [147]. Treatment with DMF attenuates astrogliosis and microgliosis of tauopathy model mice and PD model mice [148,149]. These results suggest that elimination of oxidative stress in the brain is a promising strategy for the prevention and/or alleviation of neurodegenerative diseases. Intriguingly, however, supplementation with antioxidants that quench oxidative stresses does not have any effect in AD patients [150,151]. Appropriate control of neuroinflammation, in addition to suppression of oxidative stress, appears to be necessary to conquer these diseases.

### 4.4. Aging in Skeletal Muscle and the KEAP1-NRF2 System

Aging in skeletal muscle is characterized by a gradual decline in muscle function and a reduction in muscle mass (Figure 5D). There are a spectrum of changes that occur in skeletal muscle in aged people, from physiological age-related sarcopenia to pathological muscle wasting, such as in cancer cachexia [152]. In age-related sarcopenia, muscle mass is reduced because the thickness of each muscle fiber and the total number of muscle fibers are reduced. In particular, a reduction in type II muscle fibers, which are fast fibers, is one of the characteristic features of aging [153]. When a reduction in muscle mass is combined with an increase in body fat mass, body weight remains unchanged, representing a state called sarcopenic obesity, a new category of obesity in aged people [154]. Loss of muscle mass with aging is often due to the progressive loss of motoneurons. Muscle function progressively declines because motoneuron loss is not adequately compensated by reinnervation of muscle fibers by the remaining motoneurons [155]. Mitochondrial dysfunction and impaired proteostatic mechanisms are other important contributors to the complex etiology of sarcopenia. Exercise is currently considered the only effective method to treat sarcopenia, which improves mitochondrial energetics and protein turnover [156]. Possibly related to mitochondrial dysfunction, sarcopenia patients exhibit a high blood GSSG/GSH ratio and increased plasma MDA/4-HNE protein adducts compared to nonsarcopenic patients [157]. While transiently increased oxidative stress often serves as a healthy stimulus for muscle function and regeneration [158], uncontrolled accumulation of ROS leads to pathological consequences [159]. In addition to oxidative stress, the age-associated inflammation milieu also underlies sarcopenia. Inflammation markers, including erythrocyte sedimentation rate (ESR) and C-reactive protein levels, are significantly higher in the sarcopenic group than in the nonsarcopenic group [160]. Regardless of many reports on the involvement of inflammation in sarcopenia, it is unclear whether inflammatory activation is due to aging alone or caused by comorbidities [154].

With its antioxidant and anti-inflammatory functions, NRF2 is expected to have an antiaging role in skeletal muscle. In aged *Nrf2^–/–^* mice, markers of oxidative stress, mitochondrial 4-HNE, and protein carbonyls were robustly elevated [161]. Although the absence of *Nrf2* did not impact mitochondrial content [162], mitochondrial respiratory performances were decreased [162,163] or unchanged [161] in skeletal muscles of *Nrf2^–/–^* mice compared to those in age-matched wild type mice. *Nrf2* deficiency causes a decline in skeletal muscle performance in middle-aged and aged mice, whereas minimal differences were observed in the physical performance between wild type and *Nrf2^–/–^* mice when they are young [162,164]. In contrast, the amount of muscle mass normalized to body weight is controversial in aged *Nrf2^–/–^* mice [161,164]. Because NRF2 induces a reductive cellular environment, which is rather disadvantageous for myogenesis [158], muscle mass in aged *Nrf*2^–/–^ mice may be determined by balancing the facilitation of myogenesis due to ROS accumulation and muscle wasting due to increased oxidative stress and inflammation. NRF2 is most likely enhancing skeletal muscle performance rather than exerting trophic influences on skeletal muscle.

Consistently, skeletal muscle performance measured as exercise capacity is indeed enhanced by NRF2 activation. Treatment of mice with one of the NRF2 inducers, CDDO-Im, increases their maximum running speed and distance on the treadmill compared to those treated with vehicle control [165]. Similarly, one of the NRF2-inducing phytochemicals, curcumin, improves exercise performance in mice with heart failure [166]. Moreover, genetic activation of NRF2 in skeletal muscles increases the slow oxidative muscle fiber type and improves exercise endurance capacity in female mice [167].

A seemingly common feature of aged skeletal muscles is attenuated NRF2 pathway activity. mRNA expression levels of *Nrf2* were decreased in the gastrocnemius of old wild type mice [164,168]. In myocardial cells of aged mice, nuclear translocation of NRF2 is decreased, and subsequent DNA binding of NRF2 is significantly reduced [169]. Exercise provides a clue to overcoming this issue. Exercise increases p62 phosphorylation and NRF2 activity, enhancing antioxidant protein expression [170]. Because phosphorylated p62 competes with NRF2 for KEAP1 binding [171] and because skeletal muscle-specific p62 disruption cancels out exercise-induced antioxidant gene expression [170], exercise is considered to activate the NRF2 pathway in a p62 phosphorylation-dependent manner. As described above, exercise is the only effective option for treating sarcopenia [156]. During physical exercise, reactive oxygen species are increased (reviewed in [172]). Therefore, activation of NRF2 occurs as an antioxidant response [173]. In this context, activation of the KEAP1-NRF2 system in aged muscles, which exhibits decreased NRF2 expression, may be beneficial for rapid clearance of reactive oxygen species and for enhancing the efficacy of exercise. Dietary supplementation with NRF2 inducers antagonizes age-dependent attenuation of NRF2 pathway activity. Supplementation with sulforaphane for 12 weeks restored NRF2 activity, mitochondrial function, cardiac function, exercise capacity, glucose tolerance, and activation/differentiation of skeletal muscle satellite cells in aged mice [168]. Sulforaphane also alleviates pathological conditions in muscular dystrophy model mice [174]. Thus, restoration of NRF2 activity and endogenous cytoprotective mechanisms is likely to be an effective strategy for protecting skeletal muscles from functional declines caused by aging.

## 5. Pharmacological Intervention for Increasing NRF2 Activity

To pharmacologically activate the NRF2-dependent transcription, synthetic and natural compounds are utilized. Multiple mechanistic bases are applied for achieving NRF2 activation. One is based on the KEAP1 ability to sensitively respond to electrophiles. Originally, exposure to low doses of electrophiles was found to evoke protective response from the toxicity of high doses of electrophiles, which has been called “electrophilic counterattack response” [175], and NRF2 turned out to be a key regulator of the response [10]. Electrophiles form covalent adducts to cysteine residues in the KEAP1 protein, resulting in the inactivation of KEAP1, inhibition of NRF2 ubiquitination and stabilization of NRF2. Interestingly, cysteine residues that are critical for the response to each electrophile is distinct from electrophile to electrophile, which is described as “cysteine code” (reviewed in [15]). Electrophiles are categorized into four groups according to the KEAP1 cysteine codes for NRF2 activation [176]. DMF, an approved therapeutic agent for multiple sclerosis, ameliorates the disease course and improves the preservation of myelin, axons, and neurons in an NRF2-dependent manner [177]. Phytochemicals such as isothiocyanates derived from broccoli sprouts and Japanese wasabi, carnosine from rosemary, curcumin, and sesamin are reported to activate the KEAP1-NRF2 system [178,179,180,181,182].

Another approach is disruption of KEAP1-NRF2 interaction. SQSTM1/p62 competes with NRF2 for KEAP1 binding and activates NRF2 [171]. Induction of p62 expression could result in the NRF2 pathway activation. Mimicking an action of SQSTM1/p62, small molecules that occupy an interaction surface of KEAP1 have been developed [183]. Targeting protein–protein interaction (PPI) is expected to achieve higher specificity than utilizing electrophilic reagents, because reactive cysteines in many other proteins can be conjugated with electrophiles. Still another possible approach is targeting molecules mediating KEAP1-independent NRF2 degradation pathway, such as HRD1 and IRE1 [184].

Compounds developed under these concepts are drug candidates, and some of them are now under clinical trials. For instance, sulforaphane is under phase II trials for subarachnoid hemorrhage and breast cancer, and bardoxolone methyl is under phase III trials for pulmonary hypertension and renal diseases [185]. DMF has been approved for multiple sclerosis and psoriasis [185] and expected to be effective for neurodegenerative diseases [133,149]. More detailed information of compounds and clinical trials are comprehensively described in recent review articles (reviewed in [185,186]). Supplementation with these compounds induces transient activation of cytoprotective genes and exerts beneficial effects of NRF2 including antiaging effects (Figure 2).

## 6. Concluding Remarks

To achieve a healthier and longer life, it is essential to clarify the mechanisms of the normal aging process. Although there are still many discussions and exceptions to explore, the oxidative stress theory of aging provides us with much information on normal and pathogenic processes. As described above, adequate interventions using food, drugs, physical exercise, and genetic modification decelerate aging and, as a result, ameliorate aging-related diseases. We believe that modulation of the KEAP1-NRF2 system represents a promising approach to this challenge.

## Figures and Tables

**Figure 1 antioxidants-10-01929-f001:**
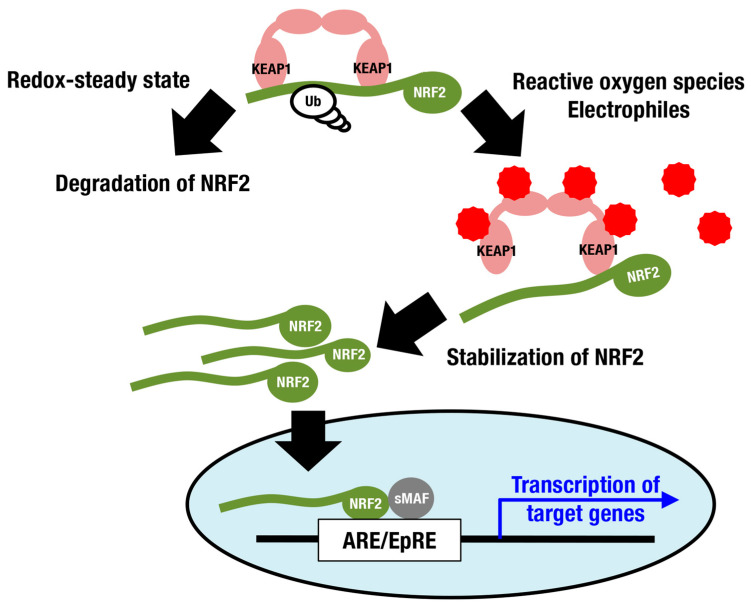
**The KEAP1-NRF2 system as a defense mechanism against oxidative stress and electrophilic stress.** Under a steady state with well-controlled redox balance, NRF2 is ubiquitinated and degraded. Reactive oxygen species and electrophiles inhibit KEAP1-dependent ubiquitination of NRF2, stabilizing NRF2 and resulting in consequent induction of NRF2 target genes. Red spiked circles indicate reactive oxygen species and electrophiles.

**Figure 2 antioxidants-10-01929-f002:**
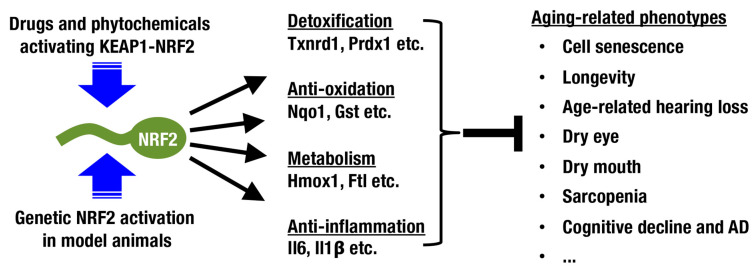
**Beneficial impacts of NRF2 activation on aging-related phenotypes.** When NRF2 is pharmacologically activated with drugs and phytochemicals or genetically activated in mice, various aging-related phenotypes are alleviated. NRF2 target genes are primarily involved in detoxification, antioxidant function, metabolism, and anti-inflammatory function.

**Figure 3 antioxidants-10-01929-f003:**
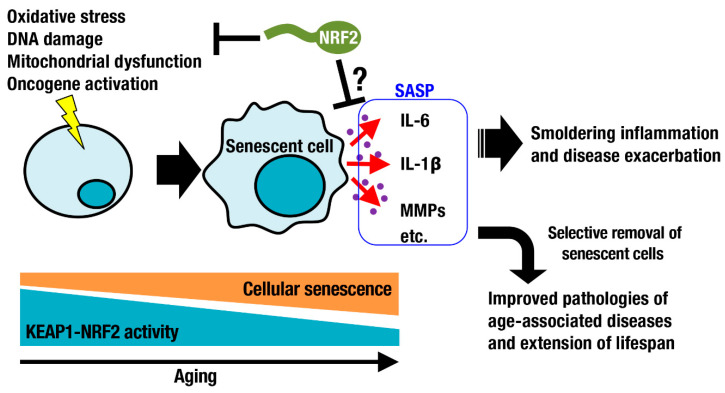
**The KEAP1-NRF2 system in cell senescence-related processes.** Sublethal damage, such as oxidative stress, DNA damage, mitochondrial dysfunction, and oncogene activation, triggers cellular senescence. During aging, senescent cells are increased in various tissues, some of which exhibit a reduction in the activity of the KEAP1-NRF2 system. Senescent cells produce inflammatory cytokines and matrix metalloproteinases, leading to smoldering inflammation and pathology. The KEAP1-NRF2 system is expected to suppress the causes of cellular senescence and SASP gene expression.

**Figure 4 antioxidants-10-01929-f004:**
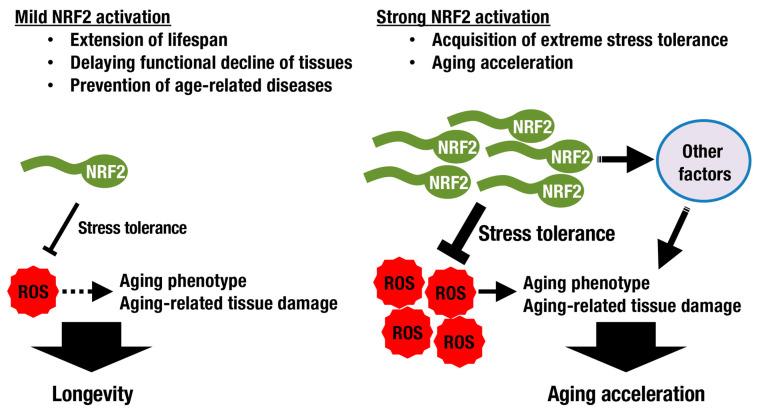
**The KEAP1-NRF2 activity and longevity.** Although ROS induce aging-related phenotypes, oxidative stress is not the only factor that regulates lifespan. In experimental models, mild activation of NRF2 extends lifespan by modulating ROS levels and attenuating aging-related phenotypes. Strong NRF2 activation, rather than conferring extreme stress tolerance, accelerates the aging process.

**Figure 5 antioxidants-10-01929-f005:**
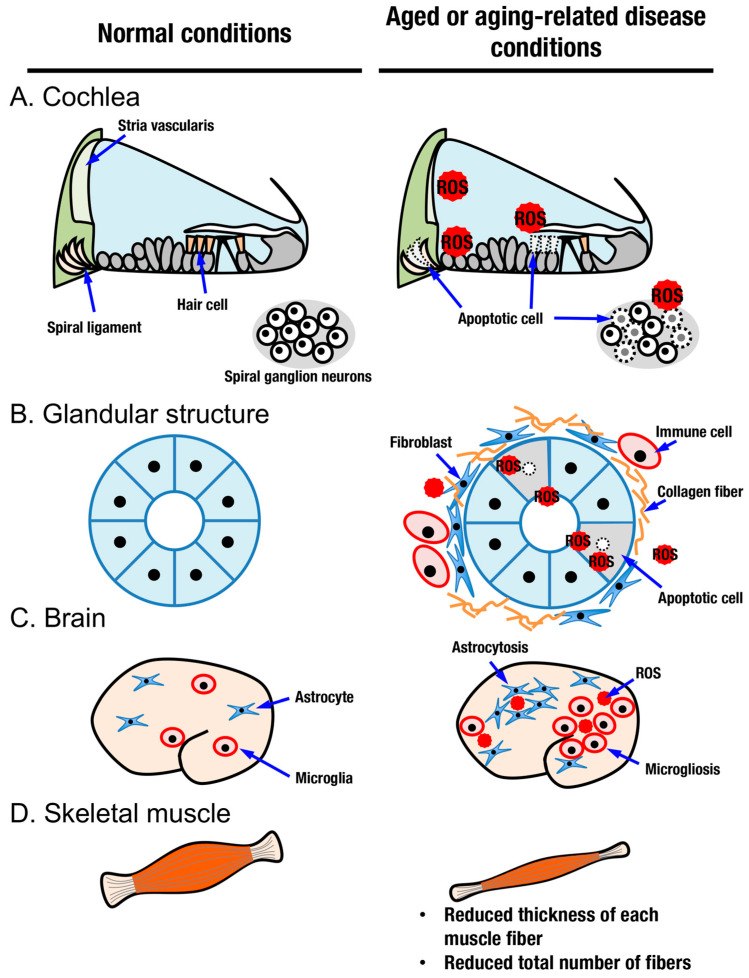
**Aged or****aging-related disease conditions in several organs.** Normal conditions and aged or aging-related disease conditions in the cochlea (**A**), glandular structures (**B**), brain (**C**), and skeletal muscle (**D**). In the cochlea, aging-related oxidative stress irreversibly impairs hair cells, spiral ganglion neurons, the spiral ligament, and the stria vascularis (**A**). In glandular structures such as the lacrimal gland and salivary gland, elevated oxidative stress, collagen deposition, immune cell infiltration, and apoptosis are observed (**B**). In aging-related neurodegenerative diseases such as AD and PD, abnormal accumulation of oxidative stress and abnormal distribution of cells termed microgliosis and astrocytosis are observed (**C**). Aged skeletal muscle exhibits reduced thickness and decreased numbers of muscle fibers (**D**).

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
