# Peer review of "The KEAP1-NRF2 System in Healthy Aging and Longevity"

_antioxidants, 2021, doi:10.3390/antiox10121929_

Round 1

Reviewer 1 Report

According to the title, this review by Matsumaru and Motohashi aims to provide a current and timely perspective about the role of the KEAP1-NRF2 system in aging. After a general introduction, the authors indicate that the main topics of the review are the contribution of the KEAP1-NRF2 system to aging-related conditions and the possibility of intervening the aging process by modulating this system (last paragraph of page 2). However, while the first part is correctly covered in the following sections, a specific section dealing with the druggability of the system is lacking. It would be highly desirable a section that summarizes the current situation of drugs able to interfere with the system, whether they are available, if they are specific, in which phase of the drug development process are they, etc. Also, if there are other potential therapeutic approaches with biologicals or gene therapy methods, they should also be covered.

Another aspect that deserves clarification is the titles given to the different sections. It is not clear the difference between cellular senescence (section 2), longevity (section 3), and tissue aging (section 4) and the KEAP1-NRF2 system. In addition, this last section (section 4) is just five lines long, so it does not make too much sense that a brief paragraph is an independent section. Then, the following sections are just an enumeration of the roles of the KEAP1-NRF2 system in different organs, and probably the review is more clear is these sections (from 5 to 8) were organized as subsections of a more general epigraph.

Author Response

According to the title, this review by Matsumaru and Motohashi aims to provide a current and timely perspective about the role of the KEAP1-NRF2 system in aging. After a general introduction, the authors indicate that the main topics of the review are the contribution of the KEAP1-NRF2 system to aging-related conditions and the possibility of intervening the aging process by modulating this system (last paragraph of page 2). However, while the first part is correctly covered in the following sections, a specific section dealing with the druggability of the system is lacking. It would be highly desirable a section that summarizes the current situation of drugs able to interfere with the system, whether they are available, if they are specific, in which phase of the drug development process are they, etc. Also, if there are other potential therapeutic approaches with biologicals or gene therapy methods, they should also be covered.

We thank the reviewer for supportive comments.

We have made an independent section (new section 5) focusing on the NRF2 activating chemicals. Recent review articles referring to the current situation of drug development have been cited. 

Another aspect that deserves clarification is the titles given to the different sections. It is not clear the difference between cellular senescence (section 2), longevity (section 3), and tissue aging (section 4) and the KEAP1-NRF2 system. In addition, this last section (section 4) is just five lines long, so it does not make too much sense that a brief paragraph is an independent section. Then, the following sections are just an enumeration of the roles of the KEAP1-NRF2 system in different organs, and probably the review is more clear is these sections (from 5 to 8) were organized as subsections of a more general epigraph.

As suggested by the reviewer, we have renumbered the section 5-8 to 4.1-4.4 making them subsections.

Reviewer 2 Report

The manuscript by Matsumaru and Motohashi reports a compilation of recent investigations into the role of Keap1-Nrf2 signaling pathway in aging, especially focusing on a number of target tissues such as sensory organs, glandular structures, brain and skeletal muscle.

The topic is a worthy subject for review, as there have been several claims that specific agents targeting the Nrf2 pathway might have important antiaging effects. The manuscript is well structured and provides updated experimental and pharmacological information which is clearly presented. I found the text excellent, although I don't feel qualified to judge about the English language. The various topics covered are relevant and the references are mostly recent and of sufficient number. The manuscript is accompanied by figures of acceptable quality that help understanding of the text.

Since the manuscript is well written, I will limit my observations to a few minor points to improve the paper.

Page 4, lines 137-141. The biological effect of the compounds capable to activate the Keap1-Nrf2 signaling pathway is essentially hormetic, i.e. ranging from beneficial to adverse effects depending on their concentration. Besides, Nrf2 overexpression increases the risk of some cancers (page 2, lines 84-86, and page 4, lines 137-141, and Figure 4) or induces drug resistance to cancer (page 6, line 189). Would it be dangerous, according to the authors, to uncritically use antiaging Nrf2 inducers with the risk of increasing susceptibility to tumorigenesis?

Page 9, lines 230-264. There is also some evidence on the role of the Keap1-Nrf2 signaling pathway in the aging of ocular tissues (Sachdeva et al. Exp Eye Res. 2014;119:111-4).

Page 12, lines 415-419. A recent finding is the relationship between the Keap1-Nrf2 signaling pathway and the p62 autophagic receptor, which links ubiquinated proteins to the autophagic machinery, enabling their degradation in the lysosome. Apparently, p62 is capable to interact with Keap1 leading to the accumulation of Nrf2 and subsequent activation of the antioxidant response (see, for example, the review by Yu and Xiao, Oxid Med Cell Longev. 2021:6635460). Do the authors think that targeting p62 with specific agents could be an efficient way of activating the Nrf2 system and ameliorating cellular senescence?

Author Response

The manuscript by Matsumaru and Motohashi reports a compilation of recent investigations into the role of Keap1-Nrf2 signaling pathway in aging, especially focusing on a number of target tissues such as sensory organs, glandular structures, brain and skeletal muscle.

The topic is a worthy subject for review, as there have been several claims that specific agents targeting the Nrf2 pathway might have important antiaging effects. The manuscript is well structured and provides updated experimental and pharmacological information which is clearly presented. I found the text excellent, although I don't feel qualified to judge about the English language. The various topics covered are relevant and the references are mostly recent and of sufficient number. The manuscript is accompanied by figures of acceptable quality that help understanding of the text.

Since the manuscript is well written, I will limit my observations to a few minor points to improve the paper.

We thank the reviewer for encouraging comments. 

Page 4, lines 137-141. The biological effect of the compounds capable to activate the Keap1-Nrf2 signaling pathway is essentially hormetic, i.e. ranging from beneficial to adverse effects depending on their concentration. Besides, Nrf2 overexpression increases the risk of some cancers (page 2, lines 84-86, and page 4, lines 137-141, and Figure 4) or induces drug resistance to cancer (page 6, line 189). Would it be dangerous, according to the authors, to uncritically use antiaging Nrf2 inducers with the risk of increasing susceptibility to tumorigenesis?

We appreciate this important comment. Based on available literatures and our observations, NRF2 inducers cause transient activation of NRF2 but not constitutive activation of NRF2, which is often observed in cancer cells with somatic mutations in KEAP1 and NRF2 genes driving malignant evolution of caners. Because the transient NRF2 activation is substantially different from persistent NRF2 activation, which is accompanied by multiple additional pathway alterations for overcoming disadvantages caused by the persistent NRF2 activation, we consider that anti-aging NRF2 inducers do not increase a risk of carcinogenesis. Rather, anti-aging NRF2 inducers are expected to strengthen anti-tumor immunity.   

Page 9, lines 230-264. There is also some evidence on the role of the Keap1-Nrf2 signaling pathway in the aging of ocular tissues (Sachdeva et al. Exp Eye Res. 2014;119:111-4).

As suggested by the reviewer, we have added the description on the ocular aging and cited the paper in the section 4.1.

Page 12, lines 415-419. A recent finding is the relationship between the Keap1-Nrf2 signaling pathway and the p62 autophagic receptor, which links ubiquinated proteins to the autophagic machinery, enabling their degradation in the lysosome. Apparently, p62 is capable to interact with Keap1 leading to the accumulation of Nrf2 and subsequent activation of the antioxidant response (see, for example, the review by Yu and Xiao, Oxid Med Cell Longev. 2021:6635460). Do the authors think that targeting p62 with specific agents could be an efficient way of activating the Nrf2 system and ameliorating cellular senescence?

We appreciate the comment. Inducing p62, which binds to KEAP1 and stabilizes NRF2, is one of the possible strategies for activating NRF2. To apply this mechanism for the treatment of aging related phenotypes and pathologies, more detailed analyses are necessary. We referred to a possible approach of inducing p62 expression for NRF2 pathway activation in new section 5.   

Round 2

Reviewer 1 Report

The manuscript can now be published in its current form.